# UserBERT: Self-supervised User Representation Learning

## Abstract

This paper extends the BERT model to user data for pretraining user representations in a self-supervised way. By viewing actions (*e.g.*, purchases and clicks) in behavior sequences (*i.e.*, usage history) in an analogous way to words in sentences, we propose methods for the tokenization, the generation of input representation vectors and a novel pretext task to enable the pretraining model to learn from its own input, omitting the burden of collecting additional data. Further, our model adopts a unified structure to simultaneously learn from long-term and short-term user behavior as well as user profiles. Extensive experiments demonstrate that the learned representations result in significant improvements when transferred to three different real-world tasks, particularly in comparison with task-specific modeling and representations obtained from multi-task learning.

## 1 Introduction

The choice of data representations, *i.e.*, how to create meaningful features, imposes tremendous impact on the performance of machine learning applications (Bengio et al., 2013). Therefore, data processing and feature engineering have been the decisive steps in developing machine learning models. To extend the applicability of the models, recent research on representation learning aims to discover the underlying explanatory factors hidden in raw data. With rapid advances in this direction, we have witnessed many breakthroughs in the areas of computer vision (CV) (Doersch et al., 2015; Sharif Razavian et al., 2014; Simo-Serra et al., 2015) and natural language processing (NLP) (Mikolov et al., 2013; Pennington et al., 2014; Lin et al., 2017).

Similarly, for building user-oriented industrial applications like next purchase prediction and recommendation, much effort has been spent on understanding business models and user behavior for creating useful features (Richardson et al., 2007; Covington et al., 2016). This is a time-consuming and application-specific process. Also, it is challenging to reuse these features or share gained knowledge between different services and applications.

To solve the issues of isolated feature engineering and task-oriented pipeline design, the pretraining-transfer learning paradigm has been explored. For example, multi-task learning (MTL) has shown promising results (Ni et al., 2018). Nevertheless, MTL has its intrinsic challenges, *e.g.*, deciding which tasks to learn jointly (Standley et al., 2019), or how to weigh tasks (Kendall et al., 2018), to achieve optimal performance. More importantly, the learning still hinges on large amounts of well-annotated user labels.

Inspired by the BERT model and its variations (Devlin et al., 2019; Lan et al., 2020), this paper explores the feasibility of understanding users in a similar way to how language is understood. We think it is conceptually intuitive to make such an analogy since understanding language and users share a similar goal, *i.e.*, understanding a conveyed message, but with different mediums. The former models what is said (sentences) while the latter learns from what is done (behavior). The syntax and semantics of a sentence are comparable with the behavioral patterns and the characteristics of a user. Hence, we hypothesize the learning procedure can be consistent in methodology as well, and propose to build upon BERT for pretraining user representations on unlabeled behavior data.

Our proposal, *UserBERT*, simultaneously learns from three categories of user data, *i.e.*, long-term and short-term behavior as well as user profiles, via a unified architecture. In particular, different action types (*e.g.*, page views, clicks and purchases) and attributes (*e.g.*, shop and item genre)

are chosen to represent long-term and short-term user behavior. For these two behavior types, we first present distinct strategies to discretize them into a sequence of *behavioral words*. Instead of modeling single user actions sequentially, the applied discretization leads to better generalization. The token representation of these behavioral words is computed by the concatenation and mean calculation of the word embeddings of the attribute IDs in each action, and this is followed by the summation of token, position and segment embeddings. These representation vectors are finally aligned with the word embeddings of user categorical profiles as the input to UserBERT. With this input, we design a novel *pretext* task, *masked multi-label classification*, and the UserBERT model is pretrained via optimizing the multi-label classifications of the multiple attributes in the masked behavioral words.

Despite the parallels between user behavior and sentences, there are substantial differences and challenges in designing the learning procedure in a coherent way. Our model is able to deal with heterogeneous user behavior data, and achieve generalization via effective tokenization and the pretraining task. While there is prior work applying BERT to task-specific user modeling (Sun et al., 2019b), this paper is built upon the assumption that behavioral patterns can be understood like the structure of a language. The UserBERT model explores integrating various types of user data in a unified architecture and learning generic representations with self-supervised signals. In our experiments, the pretrained model is fine-tuned on three different real-world tasks, and the results show that UserBERT outperforms task-specific modeling and multi-task learning based pretraining.

Our contributions are summarized as follows:

- We propose UserBERT, a self-supervised learning model, to pretrain user representations via analogizing actions in a user behavior sequence to words in sentence. It eliminates the needs of previous approaches for collecting additional user annotated labels.

- We design the discretization of user raw data sequences, the generation of the input representation and a novel pretext task for pretraining.

- UserBERT adopts a unified model architecture to enable the simultaneous learning from heterogeneous data including long, short-term behavior as well as demographics.

- We demonstrate the empirical power of UserBERT with extensive experiments. Our model is compared with task-specific models without pretraining and multi-task learning based pretraining models, and achieves performance gains on three real-world applications.

## 2    RELATED WORK

### 2.1    PRETRAINING AND TRANSFER LEARNING

Recent studies have demonstrated that pretraining on large, auxiliary datasets followed by fine-tuning on target tasks is a promising paradigm for boosting performance (Oquab et al., 2014; Donahue et al., 2014; Hendrycks et al., 2019; Ghadiyaram et al., 2019). Multi-task learning has been one of the commonly adopted approaches for pretraining due to its ability to improve generalization (Zhang & Yang, 2017; Ruder, 2017). It is shown that the pretrained MTL models can boost performance even when transferred to unseen tasks (Liu et al., 2015; Ni et al., 2018). Despite its success, MTL still has many challenges, such as negative transfer and the learning adjustment between different tasks (Guo et al., 2018). Also, MTL requires large amounts of well-annotated labels to produce satisfying outputs. There are two common forms of adaptation when transferring the pretrained models to a given target task, *i.e.*, *feature-based* in which the pretrained weights are frozen, and directly *fine-tuning* the pretrained model (Peters et al., 2019). We fine-tune pretrained models in our experiments.

### 2.2    SELF-SUPERVISED LEARNING

Deep learning models can already compete with humans on challenging tasks like semantic segmentation in the CV area (He et al., 2015) as well as a few language understanding tasks (Liu et al., 2019). However, such success relies on adequate amounts of quality training data, which can be extremely expensive or even impossible to obtain (Kolesnikov et al., 2019). As a result, a lot of

research efforts aim to liberate learning from the heavy dependency on supervised signals. Self-supervised learning (SSL), a subclass of unsupervised learning, has been drawing more attention since the recent advances in the NLP field. Instead of using supervision signals, SSL only requires unlabeled data and trains models via formulating a *pretext* learning task. There are two main types of pretext tasks: context-based (Pathak et al., 2016; Noroozi & Favaro, 2016; Sermanet et al., 2018; Wu et al., 2019) and contrastive-based (Hjelm et al., 2019; Chen et al., 2020).

## 2.3 USER MODELING

To build user-oriented machine learning applications, the key challenge is finding an expressive representation of user data so that the followed modeling can effectively extract useful information to produce good performance. For that reason, much effort has been going towards data preprocessing and transformations, such as converting user categorical attributes to embeddings and aggregating user activities like total number of visits, clicks or amount of money spent over certain time interval or a particular product genre (Richardson et al., 2007; Zhu et al., 2010). Deep learning models have successfully mitigated the dependency on human efforts due to its ability to capture underlying representations in raw data (Cheng et al., 2016; Covington et al., 2016; Zhou et al., 2018). However, these models need massive supervision signals for training, and they are mostly designed for specific tasks like recommendation (Pei et al., 2019) and click-through rate prediction (Zhou et al., 2019).

Despite the success of these deep learning models, they fail to generate promising results for real-world industrial tasks with limited labeled data. To deal with this issue, the methodology that pre-training universal user representations on massive user data, and then fine-tuning them for downstream tasks is explored. The goal is to learn a universal and effective representation for each user which can be transferred to new tasks (Ni et al., 2018). However, MTL-based pretraining still requires the collection of user labels. Also, it is limited by inherent shortcomings to achieve optimal results (Kendall et al., 2018; Guo et al., 2018). It is highly desirable for user applications to have a learning paradigm that does not require large amounts of manually annotated data. Our work is inspired by the BERT model which pretrains representations for language understanding. We aim to pretrain universal user representations by analogizing actions in a user behavior sequence to words in sentence, and apply transfer learning to downstream tasks, especially those with few labeled data, for boosting performance.

## 3 THE PROPOSED APPROACH

In this section, we first review the BERT model in brief, and then elaborate on how to extend it to user data including behavior sequences and demographic profiles.

### 3.1 THE BERT MODEL

BERT is a language representation model that pretrains deep bidirectional representations by jointly conditioning on both left and right contexts in all encoding layers (Devlin et al., 2019). The input of the BERT model is a sequence of tokens that can represent both a single text sentence and a pair of sentences. These discrete tokens consist of words and a set of special tokens: separation tokens (SEP), classifier tokens (CLS) and tokens for masking values (MASK). For a token in the sequence, its input representation is a sum of a word embedding, the embeddings for encoding position and segment.

The BERT model is pretrained with two tasks, masked language modeling (MLM) and next sentence prediction. In MLM, the input tokens are randomly masked and the BERT model is trained to reconstruct these masked tokens. In detail, a linear layer is learned to map the final output features of the masked tokens to a distribution over the vocabulary and the model is trained with a cross-entropy loss. In next sentence prediction, the inputs are two sampled sentences with a separator token SEP between them. The model learns to predict whether the second sentence is the successor of the first. A linear layer connecting the final output representations of the CLS token is trained to minimize a cross-entropy loss on binary labels. Many recent research works focus on extending the BERT model to areas beyond NLP, and successfully achieved state-of-the-art results (Sun et al., 2019a; Lu et al., 2019; Su et al., 2020; Qi et al., 2020).

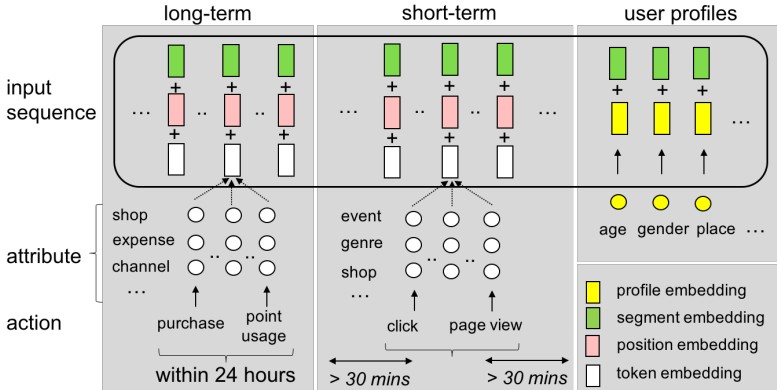

Figure 1: **Tokenization and input representation generation of long-term, short-term user behavior and profile data.** *To form behavioral words, we discretize long-term behavior into 24-hour intervals and segment short-term sequences when there is a time interval larger than 30 minutes between two actions. The word embeddings of the attribute IDs in each action are first concatenated. Then, the token representation is constructed by the mean of all action embeddings. The representation in the sequence is a summation of token embeddings and the embeddings for encoding position and segment.*

### 3.2 USERBERT

**Tokenization of user behavior sequences.** Our goal is to learn generic user representations that characterize users based on their preferences and recent interests. We decide not to sequentially model single actions in long-term and short-term user data. While such modeling is suitable for certain tasks, it is susceptible to overfitting when learning generic user representations. Instead, we learn from a sequence of clustered user actions, in which a cluster represents a routine or a spontaneous interest. Customers often make online purchases with specific intentions, *e.g.*, shopping for a shirt, cartoon books or a gift for Mother's Day. Also, many customers have long-standing preferences for particular stores and sales are heavily impacted by seasonality. These continuous or related actions form a 'word' in a behavior sequence. Similarly, we consider the same regarding short-term user behavior. Users commonly browse web content, moving between pages on an e-commerce site. During this time period, in order to capture the user's interest, we aim to estimate the theme or product genre rather than the specific order of individual actions.

Therefore, we first need to segment raw action data into a sequence of 'behavioral words' for each user, analogous to words in a sentence. In detail, we adopt different approaches for long-term and short-term data. Data representing long-standing user preferences is discretized into 24-hour intervals from 4 AM of one day to 4 AM of the next day. Short-term data is discretized if there is a time interval larger than 30 minutes between two actions, similar to the processing steps in Grbovic & Cheng (2018).

**Input representations.** In order to enable bidirectional representation learning, we transform the behavioral word sequence into a sequence of input embeddings. We first introduce the concept of *action type* and *attribute* in user actions: The action type indicates what a user does, *e.g.*, making a purchase or obtaining points for using a service, while the attribute of an action includes the shop name, the item genre and price range, etc, as shown in Figure 1. We choose different action types and attributes in our dataset to represent long-term and short-term user behavior, and propose separate tokenization strategies for them since we expect to extract inherent user preferences from regular routines over longer time periods, and short-term interests from recent, temporary interactions. In combination with demographic data, we consider the learned representations comprehensive and expressive.

To generate input representations, all attribute IDs are first mapped to fixed-length word embeddings via look-up tables. Then, the attribute embeddings of each action are concatenated. Subsequently, the token representation is constructed by the mean of all action embeddings. Finally, the input embedding vector is obtained by summing the token embeddings and the embeddings for encoding

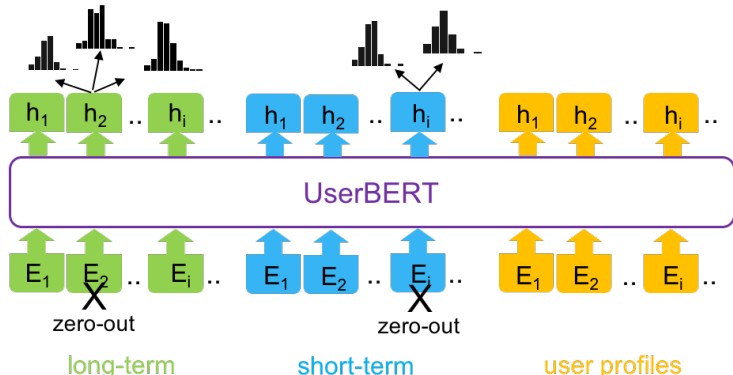

Figure 2: **Pretraining of UserBERT.** *The representation vectors in input sequences are randomly masked (zeroed-out), and then the masked input is passed through UserBERT. The model is trained to reconstruct the attributes in these masked 'words'. For each attribute, an output layer is connected to the hidden representations at the masked positions, and it is learned via minimizing the prediction errors of multi-label classifications.*

position and segment. Long and short-term user data share the same processing steps above, but each has their own definitions for token position. While the position of a token in long-term sequences is the number of days counted from the starting point of the collected training data, for short-term data it is the number of hours. The segment embedding is used to differentiate the given types of user behavior. In order to incorporate non-temporal user profile data to our modeling, we consider categorical attributes like gender as tokens in the user input sequence. For the continuous-valued attributes like age, we segment them by heuristics and convert them to categorical attributes. After mapping attributes to word embedding vectors, these are summed to the segment embedding. Note that there is no position embedding for profile embeddings since no order information needs to be captured for these user attributes. The input sequence for each user is formed by aligning the generated representation vectors of user behavior as well as the embeddings of user profiles, see Figure 1 for illustration.

**Pretraining tasks.** The generated input sequences allow us to make minimal changes to the BERT architecture and follow the practice in Devlin et al. (2019). We then pretrain our model to learn bidirectional representations. While the MLM task seems to naturally apply to our modeling, reconstructing the masked 'behavioral words' requires modification since these words contain an assembly of user actions rather than individual words used in the original BERT model. We implement *masked multi-label classification* to predict the multiple attributes in the masked behavioral words. More precisely, for each target attribute in a masked token, a linear layer is connected to the final representations and learned to map a distribution over the vocabulary of the attribute, as illustrated in Figure 2. For one masked token, the training loss is the sum of cross-entropy losses of all the attribute predictions, *e.g.*, the prediction of the shop IDs and genre IDs, etc. The final loss for one input sequence is the sum of the losses of all masked tokens.

For masking input tokens, we follow a similar process as BERT: 15% of tokens are selected uniformly, where 80% of the time the token is zeroed-out and remains unchanged otherwise. We distinguish between three segments of behavioral words from the three types of user data, *i.e.*, long-term, short-term and user profiles. For long and short-term segments, we apply the masking-prediction for pretraining our model, while we do not mask user profiles. To pretrain UserBERT, we first randomly sample a mini-batch of raw user sequences. Then, they are tokenized and transformed to input representations, which is followed by the masking step. In the end, the masked sequences are passed through the model, and the model is trained by minimizing the prediction error for reconstructing what attributes are inside the masked tokens. For each attribute type, a linear layer is learned to map the hidden representations of masked tokens to distributions over its vocabulary for conducting the multi-label classification.

Let $i$ be a randomly sampled index for masking, $w_i$ and $w_{\setminus i}$ be the masked behavioral word and the input after masking to the UserBERT. Also, let $n$ be the number of target attributes for reconstruction prediction, and $f^k(w_{\setminus i}|\theta)$ be the final output vector after softmax layer for $k$-th attribute in the masked $w_i$. The loss of the UserBERT model is:

$$L(\theta) = -\mathbb{E}_{w \sim D, i \sim \{1,..,t\}} \sum_{k=1}^{n} L_{CE}(y_i^k, f^k(w_{\setminus i}|\theta)), \tag{1}$$

where $w$ is a uniformly sampled input representation sequence from the training dataset $D$, $y_i^k$ is the ground truth binary vector for the $k$-th attribute with its corresponding vocabulary size in the masked $w_i$ and $L_{CE}$ is the cross entropy loss for the multi-label classification. Note that long-term and short-term user behavior have different types and numbers of attribute in actions. With the pretrained models, we leverage them for fine-tuning on downstream tasks.

## 4 EXPERIMENTS

We experimentally verify whether the proposed UserBERT model is able to yield generic user representations, and evaluate the performance when applying to different tasks via transfer learning.

### 4.1 DATASETS

Datasets are collected from a multitude of online ecosystem of services, including an e-commerce platform, a travel booking service, a golf booking service and others. Customers can access all services via their unique ID, and their activities across the ecosystem are linked together.

We consider two action types as long-term user behavior. The first one is the purchase action on the e-commerce platform, and the second one is the point usage history. Points are earned whenever purchases are made or when certain services are used and can be spent on any service within the ecosystem. The 'channel' attribute represents from which service users obtain points or where they spend points. We collected the purchase and point history data over a time period of three months for our experiments. For short-term behavior, we mainly focus on recent customer activities on the e-commerce website, *i.e.*, browsing and search history. The collected actions are clicks, page views and searches over a shorter time period of seven days. The detailed information on action types and attributes in the experimental data are shown in Table 1.

The user profile data is registered customer information such as age and gender. The unique number of users in the dataset is 22.5 million, the number of daily purchase and point usage samples is approximately 5 million, and the number of short-term data samples is approximately 50 million. The data is preprocessed to generate user action sequences.

Table 1: Action types and attributes in user behavior data

|  | **Action type** | **Attribute (vocabulary size)** |
|---|---|---|
| long-term | purchase, point usage | usage type (4), channel (742), expense range (17), shop (85,124), genre (11,438), hour (24) |
| short-term | click, search, page view | event type (3), shop (40,804), genre (10,386), device type (2) |

### 4.2 TARGET TASKS

We transfer pretrained models to three downstream tasks that aim to improve the customer experience. The **user targeting** task is to identify potential new customers for certain services or products, and it is formulated as a binary classification problem. The seed users who responded positively to the target service/product are positive labels, while negative ones are uniformly sampled from the rest of the users with a 3:1 ratio. The dataset is collected after the time period of the data used for pretraining. The second task, **next genre prediction**, is a multi-class prediction problem with

the aim to predict the next genre that a user will purchase from. The dataset is created from the one-month user history following the pretraining time period. The final **attribute prediction** task is predicting different user attributes such as whether a customer owns a pet. It is also a classification problem, where ground truth labels are obtained through questionnaires. The datasets of the three target tasks are split 80-20 to create training and testing datasets for fine-tuning.

## 4.3 BASELINES

The UserBERT is compared to direct modeling without pretraining and to MTL-based pretraining. The MTL models apply a multi-tower architecture in which each tower encodes one type of user data in our experiments. For the MTL-based baselines, different types of user data are passed through corresponding encoders, and the encoded representations are combined at the last layer before connecting to multiple training tasks. The dimension of the combined representations is set to 128 for all MTL models.

We collect user labels across the services in the ecosystem and pretrain MTL models with 12 multi-class classification tasks. These pretraining tasks classify the categories of user activities such as the usage frequency of certain services or attributes like type of occupation. By learning and sharing across multiple tasks, the yielded user representations are considered to be generalized and applicable for transferring to downstream tasks.

**Wide&Deep+MTL**: We generate fixed-length (1130-d) embeddings by aggregating behavior data and input them to the deep part of the model (Cheng et al., 2016). Categorical user profile data is mapped to word embeddings and concatenated before feeding it into the wide part of the model. The wide part is a linear model, while the dimensions of the 4 hidden layers for the deep side are 512, 256, 256 and 128, respectively.

**LSTM+MTL**: The same discretization and input generation is applied to long-term and short-term user behavior for this model. It is a 3-tower model, in which two LSTMs model the two types of user behavior and user profiles are modeled in the same way as the Wide&Deep model. The dimension of the hidden state in all LSTM encoders and the length limitation of both long-term and short-term data are set to 128.

**Transformer+MTL**: The architecture is the same as the LSTM+MTL model above but with two different Transformer encoders (Vaswani et al., 2017) to model long and short-term user data separately. The length of input user behavior sequence to the encoders is limited to 128 as well. We pretrain the model via minimizing the summed cross-entropy loss of the multiple training tasks.

**UserBERT**: The proposed self-supervised learning based pretraining model. It enables a simultaneous learning from long, short user behavior and user profiles. Its pretraining is done by reconstructing attributes in masked tokens via multi-label classifications.

## 4.4 EXPERIMENTAL SETUP

For UserBERT, we use the same notations as BERT, and set the number of Transformer blocks $L$ to 4, the hidden size $H$ to 128 and the number of self-attention heads $A$ to 4. The input sequence length of both long-term and short-term data is limited to 128 in the experiments. For fair comparison, we pretrain all models using the Adam optimizer with a learning rate of $1e\text{-}4$ and a batch size of 16. We fine-tune models using the same learning rate and a batch size of 128. Pretraining of 400K batches of the UserBERT model takes approximately 12 hours using our PyTorch implementation, run on two GeForce RTX 2080 Ti GPUs.

For fine-tuning each target task, the combined encoder representations of the MTL-based models are fed to an output layer, while the fine-tuning of UserBERT is done by connecting the hidden representations of the first token to an output layer for each task. After plugging in task-specific inputs and outputs, we fine-tune the parameters of pretrained models in an end-to-end way.

## 4.5 EXPERIMENT RESULTS

**User Targeting.** We show the results for two different services. The sizes of the datasets are 30,204 samples and 31,106, respectively. Compared to the size of the pretraining dataset, the use cases of

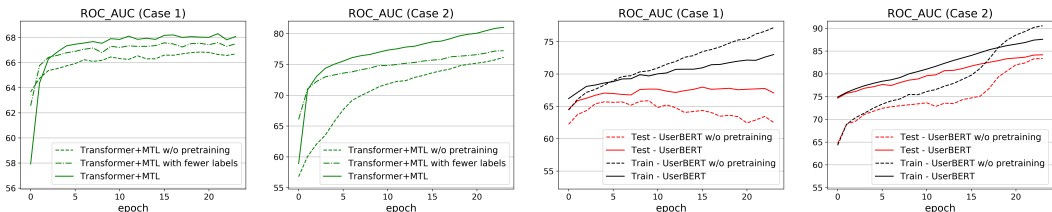

Figure 3: **Fine-tuning performance comparison for the *user targeting* task.** *ROC_AUC and accuracy results on two use cases, predicting new users for two different services.*

Figure 4: ROC_AUC comparison between Transformer-based MTL models with different numbers of labeled data.

Figure 5: Performance comparison between UserBERT with and without pretraining on *user targeting* task

this task only have few labeled data. Classification performance in terms of accuracy and ROC_AUC are shown in Figure 3. The LSTM model, which sequentially models user behavior, has relatively low accuracy. One possible explanation is that the sequential order of user actions does not provide useful information for this task. From our experience the user targeting task focuses on patterns from relatively static user preferences. The Wide&Deep model shows competitive performance, which is reasonable since our exploratory analysis indicates that user profiles are important features. The performance of the Transformer-based models reveal that the underlying explanatory factors for this task can be captured by attention networks. UserBERT outperforms other models in both use cases by a substantial margin. We hypothesize that, compared to Transformer-based MTL, the learning of the UserBERT is not limited by the multiple training tasks and is able to learn more expressive and generic representations from the input.

To further demonstrate the advantage of the proposed method over MTL-based pretraining, we pretrain Transformer-based MTL models with different numbers of labels before fine-tuning. We evaluate three models: without pretraining, trained on 30% of the available labels and trained using all labels. The comparison indicates that the performance of MTL is significantly affected by the number of training samples. As shown in Figure 4, more annotated training data contributes to performance gain. The model without the pretraining step shows the worst performance. In contrast, the pretraining of the UserBERT does not require the additional collection of supervision signals, and therefore is not impacted by either the quantity or the quality of user annotations.

We also directly apply UserBERT to these two use cases without pretraining to verify whether the user targeting task benefits from the pretraining step. The ROC_AUC comparison between UserBERT with and without pretraining is shown in Figure 5. The pretrained models outperform the direct modeling significantly. This indicates that the pretraining step can extract useful information and enables the followed fine-tuning to boost performance for downstream tasks. From the error curves during training, we also observe that models tend to overfit quickly without pretraining. The pretrained UserBERT model achieves more generic user representations and yields significant accuracy improvements when adapted to new downstream tasks.

**Next Genre Prediction.** The test dataset contains 586,130 users, and we run 10 epochs of fine-tuning for each pretrained model. The mean average precision (mAP) comparison is shown in Table 2. The UserBERT model outperforms baseline models by a large margin. This task requires understanding of both long-term preferences as well as recent interests of customers. Prediction models should be able to pick out candidate genres from user habits over a longer time range,

Table 2: mAP@10 comparison after 10-epoch fine-tuning on *next genre prediction* task.

| Model | mAP(%) |
|---|---|
| Wide&Deep+MTL | 7.65 |
| LSTM+MTL | 6.90 |
| Transformer+MTL | 7.10 |
| UserBERT | **8.97** |

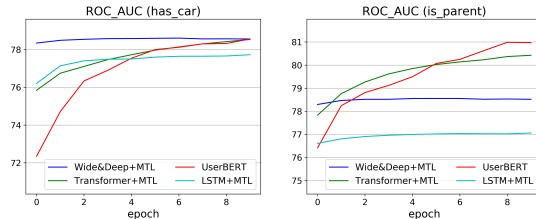

Figure 6: ROC_AUC comparison on *attribute prediction* task.

and then identify likely ones as prediction results from latest interest trends. More specifically, a model should understand how users typically use services in the ecosystem as well as what they are currently interested in. The architecture of the baseline models learns from different types of user data separately and combines the last-layer representations for training. It fails to sufficiently capture the correlations. On the contrary, UserBERT benefits from the unified structure of the user data and captures more accurate correlations, not only within certain types of user behavior, but also between different behavior types via attention networks.

Since it is common that users make purchases from only a subset of genres, we also built an intuitive but strong baseline that sets predictions as the most popular genres ranked in descending order by the total number of purchases, and compared it against all pretrained models. The mAP@10 is 4.22%, demonstrating the effectiveness of the pretrained models.

**Attribute Prediction.** In general, it is challenging to predict user attributes because the predictive signals in the behavior data are very sparse. In other words, the target user attributes may not be strongly correlated to behavior data. Therefore, this prediction task evaluates the model's ability to discover hidden explanatory factors in the raw data. We show experimental results of two use cases: one is to predict whether a user has a car while the other one is to predict if a user is a parent. These two tasks are denoted as *has_car* and *is_parent*.

The dataset for the *has_car* task contains 448,501 samples and the one for the *is_parent* task contains 400,268. The classification results of 10-epoch fine-tuning are shown in Figure 6. From the *has_car* results, we observe that the Wide&Deep model shows good performance, although other models eventually reach similar accuracy. We believe this is due to the fact that user demographics like age and living area are important features for this task. It seems challenging for models to extract other decisive patterns from either long-term or short-term user behavior. On the other hand, whether a user is a parent or not seems to present different characteristics in terms of how they behave on an e-commerce or travel booking platform. These patterns can be captured by deep learning models like UserBERT and Transformer-based models. UserBERT is able to match and eventually outperform the baseline models.

## 5 CONCLUSIONS

This paper introduces a novel paradigm to understand user behavior by using the analogy to language understanding. We present UserBERT, an extension of the BERT model to user data, for pretraining user representations in a self-supervised way. It explores and demonstrates the possibility for user-oriented machine learning tasks to alleviate the dependency on large annotated datasets. Extensive experiments show that a well-designed pretrained model with self-supervision is able to outperform fully supervised learning models when transferred to downstream applications.

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
