# OpenReview forum: "UserBERT: Self-supervised User Representation Learning"
_ICLR.cc/2021/Conference — Reject_

### Official Review · AnonReviewer3 · 2020-10-27
**Interesting approach to model user behavior but lacking of modeling details**

**Rating:** 5
**Confidence:** 5

**Review:**

Clarity:

I find the paper lacks details on multiple aspects (please see my comments below).

Originality:

The idea introduced in the paper is interesting. Most of the originality comes from the fact that the author is converting a user modeling problem into a pre-training problem defined on a special "vocabulary", aka action types, attribute IDs.

Significance:

I am not convinced that the result is significant. Mostly because the experiment is hard to be reproduced due to lack of description of modeling details and the dataset being used.


Some other comments:

* It would be more helpful to point out the difference in more details that how your work is different from other related work in section 2.1 and 2.2.

* In section 3.2, "Tokenization of user behavior sequences", if I understand correctly, (at least some subset of) the attribute type of an user action is largely dependent on an external topic model or topic classifier (for example: to identity the main topic of a webpage, "sport news", or to recognize the relevant entity from a shop, "nike shoes" ). This part is not clearly explained in the paper. I would recommend using an appendix section to explain it further. Otherwise it is impossible for other readers to reproduce the results mentioned in this paper.

* Can you rephrase this sentence "The token representation is computed by the concatenation and mean calculation of the word embeddings of the attribute IDs in each action" (Under Figure 1.) ? I am confused how this is done. Is it a mean word embedding for each attribute ID? Then we concatenate all of the mean embedding vectors?

* In section 3.2 "Input representations", could you give an example of "token" (I see you said "age" is a token, is there any other examples? is "token" something only related to user demography) and "action"? what is the difference between them? This give me confusion when reading the second paragraph under "Input representations".

* Under "Pretraining tasks", "The final loss for one input sequence is the weighted sum of the losses of all masked tokens.". Can you elaborate on what the "weighted sum" refers to? What is the weight? Are you assigning some weights to different type of user behavior, e.g., long-term, short-term and user profiles? How is that weight being decided?

* In the experiment section 4.1, the dataset used in the paper does not seem public-available. Or do I missing something? This also blocks reproducibility of  the paper results. Is there anything you can use in the public world? Otherwise can you at least give more details (besides those already in 4.1) on the underline dataset in  the appendix?

---

> ### Author Response · Authors · 2020-11-24
> **Responses to AnonReviewer3. We appreciate your time and feedback.**
>
> We value your time and thank you for the helpful questions and comments. Please kindly find our following responses.
>
> Q: It would be more helpful to point out the difference in more details that how your work is different from other related work in section 2.1 and 2.2.
>
> --> A: We totally agree with your comment. Please refer to our comment: paper update #1 for the modifications.
>
> Q: In section 3.2, "Tokenization of user behavior sequences", if I understand correctly ...
>
> --> A: We applied a practical way for tokenization based on the characteristics of our dataset. As described in Figure 1, the long-term user behavior (i.e. purchase and point usage) is discretized into 24-hour intervals, and short-term data (i.e. click, search and page view) is segmented when there is a time interval larger than 30 minutes between two actions.
>
> Q: Can you rephrase this sentence "The token representation ...
>
> --> A: Correct. We modified the captions of Figure 1 accordingly.
>
> Q: In section 3.2 "Input representations", could you give an example of "token" ...
>
> --> A: The ‘token’ is not just related to user demography. Instead, it is derived from long-term and short-term user behavior as well. We propose to analogize the ‘behavioral words’ of user behavior to the ‘token’ of a language, so that it becomes feasible to apply the BERT model to user data. An example of a token of short-term user behavior can be a series of clicks on an e-commerce site, which reflects the user interest during that time period.
>
> Q: Under "Pretraining tasks", Are you assigning some weights to different type of user behavior ...
>
> --> A: The masked tokens of the three different types of user data are weighted equally in our experiments. We made modifications accordingly in our manuscript for better clarity.
>
> Q: In the experiment section 4.1, the dataset used in the paper does not seem public-available ...
>
> --> A: We fully agree that it is beneficial for our work to show experimental results on public datasets, and we are working on this part. Please also refer to the second paragraph of our comment: paper update #1.

---

### Official Review · AnonReviewer1 · 2020-10-28
**Limited significance and weak experiments**

**Rating:** 4
**Confidence:** 3

**Review:**

##
I have read the author response and I still think the paper is limited in terms of novelty, significance and experiments. I would like to keep my current score.
##

This work proposed a self-supervised pre-training approach to model users from user behavior time-series data. Given time-series data of user actions on ecommerce websites, web browsing, etc.,  a self-supervised prediction task similar to masked language modeling is introduced where a transformer model predicts masked actions from the context. Paper claims to show that the proposed pre-training approach performs better than baselines pre-trained using multitasking on user behavior prediction tasks such as next purchase.

Pros
* Good user models can be useful in recommendation systems and paper tackles an important problem
* Proposed pre-training approach based on time-series data is interesting and looks reasonable
* Paper is generally well written and easy to follow

Cons
* Limited novelty and significance
* Weak experiments
* Related work on user modeling may not be comprehensive enough

Related work not comprehensive: The paper doesn’t do a good job of positioning the work in the context of prior work. I don’t see any papers discussed in the user modeling section of the related work, which raises concerns about the significance of this work.

Poor baselines: The model is compared against baselines constructed by the authors, some of which are vaguely described (Eg. I don’t understand the Wide&Deep baseline). I am not sure if these are strong baselines. It is also unclear how good the multi-task learning baselines are and there isn’t any significant discussion about the tasks/hyperparameters used to train these baselines.

Limited novelty/contributions: The approach can be considered an application of masked language modeling. I’m not sure if the details about tokenization bear much significance. If the authors claim this to be a contribution, the same should be verified by comparing against other ways of tokenizing the data. Modulo this detail, the approach is a straightforward application of BERT-style pre-training.

I don’t think the analogy between language and user behavior needs to be mentioned/adds any value. Masked language model pre-training can be applied to any time-series data.

Although the paper claims to model users, I couldn’t find details in the paper about how exactly a user-model is constructed from the pre-trained model and how these models are fine-tuned on the target task.

While the paper attempts to address an important problem, it has serious issues in terms of limited novelty/significance of contributions and weak experiments.

---

> ### Author Response · Authors · 2020-11-24
> **Responses to AnonReviewer1. Thank you so much for the questions and comments.**
>
> The questions and comments are much appreciated for improving our manuscript. We respond to questions and comments as follows.
>
> Q: Related work not comprehensive
>
> --> A: We modified section 2.3 for a thorough survey. Please refer to our comment titled as ‘paper update #1’ for more details.
>
> Q: Regarding baselines
>
> --> A: We’d like to emphasize that although the baselines are implemented by authors, they are commonly applied models for user modeling in industry. We added more details for each baseline model. Please kindly refer to Section 4.3. Wide&Deep model does not model user behavior as sequential data. Instead, it represents a traditionally applied approach, which aggregates user behavior to generate fixed-length embeddings and then input them a feedforward neural network[1]. Such modeling is one of the most intuitive approaches and widely adopted for many applications.
>
> Q: I couldn’t find details in the paper about how exactly a user-model is constructed…
>
> -->A: We added a paragraph in section 4.4 to explain how the finetuning of MTL-based pretrained models and UserBERT is done.
>
> Q: Regarding the comparison between different tokenizations
>
> --> A: Thank you for the comment. We completely agree that different tokenizations should be compared. We’d like to add such comparisons in the final version.
>
> [1]: Paul Covington, Jay Adams, and Emre Sargin. Deep neural networks for youtube recommendations. In Proceedings of the 10th ACM Conference on Recommender Systems, pp. 191–198, 2016.

---

### Official Review · AnonReviewer4 · 2020-10-28
**Official Blind Review**

**Rating:** 3
**Confidence:** 5

**Review:**

UserBERT: Self-supervised User Representation Learning

######################################################################

Summary:

The paper provides an extension of BERT to user data for pre-training user representation in a self-supervised manner. In particular, it analogise the user behaviour sequence to words in a sentence and leverages the Masked Language Model (MLM) approach typically used in NLP to train the user embedding. To facilitate such extension, the paper also proposes a discretisation approach and a unified input structuring to include long-term, short-term and demographic information.

######################################################################

Pros:

1. Even though, the idea of extending the self-supervised pre-training for user representations is not new, it is still an interesting area of research

2. The discretisation of user behaviour signals over long-term and short-term to form “behavioural words” is quite reasonable

#####################################################################
Concerns:

The key concerns about the contributions of the paper are as follows:

Overall, the novelty of this work is very limited. To elaborate:
The major contribution of this work is two fold:
- Discretisation of raw user behaviour sequences (for long-term and short-term) and
- Using the discretised aggregated “behaviour words” as inputs to the BERT architecture as is.

The other claimed contributions such as having a unified architecture and experiments to validate the approach do not seem substantial.

When it comes to “discretisation”, though the idea seems appropriate, two crucial questions regarding this step are not validated:

1) There is no empirical evidence presented in the paper which shows “discretisation” improves UserBERT’s accuracy. Since it is a major contribution, I request the authors to design and implement an ablation study to address this point.

2) Seemingly, the authors have come up with a hand crafted/heuristic-driven  approach for discretisation. Why can’t it be data-driven too? Meaning, can clustering of actions be done in a data-driven manner? If so, what is the difference in accuracy between the proposed heuristic-driven and the data-driven alternative.

When it comes to the second contribution, it is certainly not novel i.e., the paper does not propose any architectural change to BERT. Though the paper claims that the presented model is a unified model to learn long term, short term and demographics based user profile, the unification is brought upon as a by-product of feeding multimodal inputs to vanilla BERT.

Hence, the overall novelty of the paper is very limited.


The following comments are my major concerns in each section of the paper:

Section 2:

While the authors have reviewed some literature in transfer learning and self supervised learning and have cited some relevant work, they have not cited even one reference in section 2.3 which on “User Modeling” i.e., the main theme of the paper. I request the author to make a thorough survey and cite related work in section 2.3 and also highlight how UserBERT is different from them.

Section 3:

Overall, this section (and section 4) lack cohesion and can be written clearer with the figures, tables, algorithms and descriptions. This could help the reader better understand the approach. For instance, the following main points in the approach are not explained well:

1) The paper states “the final loss for one input sequence is the weighted sum of the losses of all masked tokens”. There is no detail what the weights are, whether they are assigned based on heuristic or learnt.

2) The approach considers “ordinal attributes” such as expense and age similar to “categorical attributes” (e.g., each age has a unique embedding). This seems counterintuitive and there is no empirical evidence to show that this counterintuitive design works well.

3) It is not clear how to use the hidden representation to predict attributes from the transformed masked tokens. More precisely, it is possible that many attributes belonging to different actions are masked and then converted into one token embedding. So what attributes are to be predicted in the final fully connected layer, in this case?

4) Minor concern: what does E stand for in equation 1?

Section 4:

Overall, in this section, the experimental design is not comprehensive, and the results are not convincing for the following reasons:

- Along with the Wide&Deep, LSTM, Transformers as baseline, it would have been better to also include vanilla BERT to the baselines against which the UserBERT can be compared. In fact, Vanilla BERT would be the closest and most appropriate baseline for comparison. Hence, I request the authors to include it.

- All the experiments are conducted on custom datasets. Since user profiling is an extremely useful and ubiquitous activity that benefits multiple domains, I request the authors to experiment UserBERT on well-known open source e-commerce (and other user profiling) datasets (ref: [1] and [2]). In fact, the profiles could be tested on downstream tasks like “next genre prediction” with these datasets. This will help the reader to trust the UserBERT model better.

- Input representation, being one of the major contributions of the paper, it would give more insights if an ablation study is made on the user behaviour data (long-term features, short-term features, demographic features) to compare and contrast the contribution and lift by each of the behaviour categories

- In the attribute prediction task, within the two attributes experimented, the performance of the proposed model is quite unconvincing. Would benefit if more experiments are performed.

- For the “next genre prediction”, though there are more than 10k genres, each users’ typically have a very small subset of interest. Therefore, it would be more informative if can compare the model’s mAP@10 with the user-level mode’s mAP@10.

- The discussions of results are very vague and could be a lot deeper and precise.


#####################################################################

Questions during rebuttal period:

Please address and clarify the concerns above

#####################################################################

Reference

[1] Sun, Fei, et al. "BERT4Rec: Sequential recommendation with bidirectional encoder representations from transformer." Proceedings of the 28th ACM International Conference on Information and Knowledge Management. 2019.

[2] Kang, Wang-Cheng, and Julian McAuley. "Self-attentive sequential recommendation." 2018 IEEE International Conference on Data Mining (ICDM). IEEE, 2018.

---

> ### Author Response · Authors · 2020-11-24
> **Responses to AnonReviewer4. Thank you very much for the questions and comments.**
>
> We’d like to express our appreciations for the insightful comments to help polishing our work. Please find our responses as follows.
>
> Q: Regarding the section 2.3 on ‘User Modeling’
>
> --> A: We completely agree that related work on user modeling should be cited and how our work is different should be made clear. Please refer to our comment : paper update #1 and confirm our modifications of section 2.3.
>
> Q: The paper states “the final loss for one input sequence is the weighted sum of the losses of all masked tokens” ...
>
> --> A: Thank you for the comment. The current weighting is heuristic-based, and different types of user data are actually weighted equally. We modify the manuscript accordingly.
>
> Q: The approach considers “ordinal attributes” such as expense and age similar to “categorical attributes” ...
>
> --> A: We’d like to clarify that not each ordinal value of these attributes has a unique embedding. In fact, we discretize the ordinal attributes and convert them to ‘categorical attributes’. We admit that the processing needs to be made clear, and we add more explanations in the ‘input representations’ of section 3.2.
>
> Q: It is not clear how to use the hidden representation to predict attributes from the transformed masked tokens..
>
> --> A: Thank you for the question. For a masked token, UserBERT makes predictions for all the attributes of different actions. It is a multi-label classification for every attribute. Therefore, for each attribute there is an output layer connecting to the hidden representation of the masked token. For one masked token, the loss is a sum of all the losses of multiple multi-label classifications for all attributes.
> To make it clear, we added more details in the captions of Figure 2 and the ‘pretraining tasks’ of section 3.2. Also, We corrected equation 1 as mentioned in our ‘paper update #1’
>
> Q: Minor concern: what does E stand for in equation 1?
>
> --> A: E in equation 1 stands for ‘expectation’.
>
> Q: ... it would have been better to also include vanilla BERT to the baselines against which the UserBERT ...
>
> --> A: We totally agree that Vanilla BERT on original user behavior sequences is another way to learn user representations. We’d like to add it in the final version of this work.
>
> Q: All the experiments are conducted on custom datasets ...
>
> --> A: Please kindly refer to the ‘paper update #1’ for our detailed responses.
>
> Q: ...it would give more insights if an ablation study is made on the user behaviour data ...
>
> --> A: We do agree that it brings more insights to have a comparison on the different types of user data. We already had some initial results on next genre prediction. Only using user profiles gives a mAP of 6.8%, while the mAP of applying user sequential behavior data is 8.5%. Complete usage of all available data yields a result of 8.97%, as shown in Table 2. We plan to run more experiments on more use cases of all target tasks.
>
> Q: In the attribute prediction task, within the two attributes experimented, the performance of the proposed model is quite unconvincing. Would benefit if more experiments are performed.
>
> --> A: We agree more experimental results would be beneficial and will try to expand UserBERT to more use cases.
>
> Q: the discussions of results are not deep enough
>
> --> A: Thank you very much for the comment.

---

### Official Review · AnonReviewer2 · 2020-10-28

**Rating:** 4
**Confidence:** 4

**Review:**

The paper presents an approach to learning user representations based on activity patterns on e-commerce websites and a user profile. The method turns activity patterns into a sequence of discrete tokens based on the action type and attributes that correspond to a certain action. A self-supervised transformer is trained on this data with a masked language modeling (MLM) objective. Data is compartmentalized as long-term patterns such as a purchase or the use of reward points or short-term such as clickthrough data or user profile information such as user age, gender, or location. Separate segment and position embeddings are used within each compartment. Since each masked token is a high-level action type that may have many attributes, predicting a masked-token is cast as a multi-label classification problem over attributes.

The trained model is evaluated on downstream user targeting and user attribute prediction benchmarks.

Overall, the paper is a straightforward application of BERT pre-training (with MLM only) to learn user representations. The main contribution of this work is the tokenization/discretization strategy and multi-label classification to enable masked language modeling.

Strengths

The overall approach of leveraging user activity patterns to learn user embeddings in a self-supervised way is well motivated and uses well-established methods like BERT to achieve this goal.
The discretization/tokenization and multi-label classification idea is well thought out.

Weaknesses

Limited discussion of previous work in this space of user representation learning (the user modeling section lacks any citations)
Straightforward BERT application with somewhat mixed results on some tasks (case 1 on user targeting and “has_car” attribute prediction)
It is hard to determine what improvements are meaningful since the datasets used are standard to the best of my knowledge and there aren’t any significance testing.

Questions & Comments:

What is the baseline accuracy of predicting the same genre as the previous genre purchased in the next genre prediction task?

The limited supervision setting was unclear to me (especially in Figures 4 & 5). Please correct me if I’ve interpreted the results incorrectly. Figures 4 & 5 investigate *supervised* pre-training on labeled user data. The Transformer + MTL model when pre-trained on 12 classification tasks that require supervised data 1) gets better with more supervised pre-training data 2) does better than UserBERT with the same pre-training on supervised tasks. Figure 5 shows that UserBERT begins to overfit.

A motivation mentioned was that self-supervised pre-training allows for strong semi-supervised learning and learning with limited supervision however most performance curves report performance as a function of the number of epochs rather than the amount of labeled data available for downstream tasks like user targeting and attribute prediction. Strong performance with limited labeled data would make this work better.

The paragraph that starts with “Inspired by the BERT model and its variations” contains a sentence like “understanding users in a similar way to how language is understood” and “syntax and semantics of sentence are comparable with behavioral patterns and characteristics of a user” both (especially the latter) are pretty bold claims and are largely unnecessary to motivate the proposed model. It should suffice to say that the model is inspired by BERT which has been immensely useful across a host of NLP tasks.

The section on user modeling has no references to previous work that looks at building user representations. The recommendation systems literature contains several such pieces of work and these should be discussed. Along these lines, how would the learned representations compare to those obtained from say SVD of the user-purchase/user-click/user-search matrix?

This work appears to have some potential for misuse, especially when trying to infer protected user attributes from user embeddings. This deserves to be discussed in some detail.

It's not clear to me why reporting area under the ROC curve as a function of the number of epochs is meaningful. Why aren’t models trained with early stopping?

---

> ### Author Response · Authors · 2020-11-24
> **Responses to AnonReviewer2. Thank you so much for the valuable feedback.**
>
> The authors would like to thank the reviewer for your time and helpful comments.
> We respond questions and comments as follows:
>
> Q: What is the baseline accuracy of predicting the same genre as the previous genre purchased in the next genre prediction task?
>
> --> A: We do not have the results for predicting the same genre as the previously purchased item.
> Although it does exist that customers purchase from the same genre as the previously bought item in our dataset, it is more common for users to buy from popular genres. In fact, we compared against a baseline of predicting the next purchase as the most popular genres. The mAP@10 is 4.22%. We think this is a stronger baseline than predicting the same genre. We added a paragraph for  the description of this baseline and its result in section 4.5.
>
>
> Q: Regarding the experimental results shown in Figure 4 and Figure 5...
>
> --> A: Figure 4 and 5 are for different purposes. Figure 4 is to demonstrate that the performance of MTL-based pretraining is easily affected by the number of available labels. It does get better with more supervised signals, but limited labels fail to generate comparable results. Please note that the Transformer+MTL is not better than UserBERT.
> Figure 5 is to demonstrate the advantage of pretraining for our proposal. It shows that UserBERT with pretraining consistently outperforms the direct modeling without pretraining. Also, the model without pretraining overfits quickly.
>
>
> Q: Regarding showing experimental results as a function of the amount of labeled data
>
> --> A: We totally agree that showing experiment results as a function of the amount of labeled data can help to present our proposal better. But we’d like to mention that the three target tasks used in our experiments are already in the setting of only having few available labeled data, especially the first task: user targeting. The experimental results on these three target tasks already demonstrated that UserBERT outperforms baselines with different numbers of limited labeled samples.
>
> Q: The section on user modeling has no references to previous work …
>
> --> A: We fully agree and address this by modifying section 2.3. Please refer to our comment: paper update #1.
>
> Q: It's not clear to me why reporting area under the ROC curve as a function of the number of epochs is meaningful. Why aren’t models trained with early stopping?
>
> --> A: We think reporting different metrics as a function of the number of epochs can present a comprehensive image to the audience regarding how the training/learning goes. We’d like to point out that it is a common way to show results with respect to the learning iterations/epochs [1][2].
> Early stopping and then reporting the performance is suitable for sure, but we think it is also a valid way to present experimental results as a function of the number of training iterations.
>
> [1] Guorui Zhou, N. Mou, Ying Fan, Qi Pi, Weijie Bian, C. Zhou, Xiaoqiang Zhu, and K. Gai.  Deep interest evolution network for click-through rate prediction. In AAAI, 2019.
>
> [2] Yabo Ni, Dan Ou, Shichen Liu, Xiang Li, Wenwu Ou, Anxiang Zeng, and Luo Si.  Perceive your users in depth: Learning universal user representations from multiple e-commerce tasks.  In Proceedings of the 24th ACM SIGKDD International Conference on Knowledge Discovery & Data Mining, pp. 596–605, 2018.

---

### Author Response · Authors · 2020-11-24
**Paper update #1**

The authors sincerely thank the reviewers for the insightful comments to make our work better. We’d like to respond to common comments and clarify major updates here.

We agree that a thorough discussion on previous user modeling research positions our work more clearly, and addressed this common comment with a modified section 2.3. We make our motivation clear by pointing out that deep learning models with huge amounts of labeled data may not work for a common scenario in real-world applications, in which only limited data can be collected and available for model training.

We made corrections on Equation 1. We admit it may have caused confusions on how the attributes in masked tokens are predicted. Also, we added more explanations in ‘pretraining tasks’ of section 3.2 to better present the pretraining process.

We completely agree that experiments on public datasets demonstrate the advantages of the proposed UserBERT better, and we will add experimental results in the final version of this work. But we’d like to also emphasize that our work is motivated by real-world business cases on a company dataset. Our experiments contributed to the improvement of different tasks/applications in practice. For user modeling, it is able to bring a new perspective for learning universal and general user representations. Since our proposal does not require the collection of additional user labels for training, it has its significant practicality and contributes much in terms of saving cost and protecting privacy.

---

### Decision · Program_Chairs · 2021-01-07
**Final Decision**

**Decision:**

Reject

**Comment:**

The paper discusses an extension of BERT for learning user representations based on activity patterns in a self-supervised setting. All reviewers have concerns about the validity of the claims and the significance of the experimental results. Overall, I agree with the reviewers that the paper needs more work to be published at ICLR. I recommend rejection.